# Epidemiology and Risk Prediction Model of Multidrug-Resistant Organism Infections After Liver Transplant Recipients: A Single-Center Cohort Study

**DOI:** 10.3390/bioengineering12040417

**Published:** 2025-04-14

**Authors:** Chuanlin Chen, Desheng Li, Zhengdon Zhou, Qinghua Guan, Bo Sheng, Yongfang Hu, Zhenyu Zhang

**Affiliations:** 1School of Clinical Medicine, Qinghai University, Xining 810000, China; 15003680407@163.com (C.C.);; 2Department of Liver ICU, Beijing Tsinghua Changgung Hospital, School of Clinical Medicine, Tsinghua Medicine, Tsinghua University, Beijing 100000, China; 3Department of Clinical Pharmacy, Beijing Tsinghua Changgung Hospital, School of Clinical Medicine, Tsinghua Medicine, Tsinghua University, Beijing 100000, China

**Keywords:** liver transplantation, multi-resistant organisms, postoperative infection, nomogram

## Abstract

**Objective:** Accurate risk stratification at an early stage may reduce the incidence of infection and improve the survival rate of recipients by adopting targeted interventions. This study aimed to develop a nomogram to predict the risk of multidrug-resistant organism (MDRO) infections in liver transplant (LT) recipients. **Methods:** We retrospectively collected clinical data from 301 LT recipients and randomly divided them into a training set (210 cases) and validation set (91 cases) using a 7:3 split ratio. Factors related to the risk of MDRO infection after LT were determined using univariate and multivariate bidirectional stepwise logistic regression. The model’s predictive performance and discrimination ability were evaluated using receiver operating characteristic (ROC) curves, calibration curves, and decision curve analysis (DCA). **Results:** 56 (18.60%) patients developed a MDRO infection, including 37 (17.62%) in the training cohort and 19 (20.88%) in the validation cohort. Ultimately, five factors related to MDRO infection after LT surgery were established: ascites (OR = 3.48, 95% CI [1.33–9.14], *p* = 0.011), total bilirubin (OR = 1.01, 95% CI [1.01–1.01], *p* < 0.001), albumin (OR = 0.85, 95% CI [0.75–0.96], *p* = 0.010), history of preoperative ICU stay (OR = 1.09, 95% CI [1.01–1.17], *p* = 0.009), and length of ICU stay (OR = 3.70, 95% CI [1.39–9.84], *p* = 0.019). The model demonstrated strong discrimination, and the area under the curve (AUC), sensitivity, and specificity of the training set were 0.88 (95% CI [0.81–0.94]), 0.82 (95% CI [0.76–0.87]), and 0.86 (95% CI [0.75–0.98]), respectively, while for the validation set, they were 0.77 (95% CI [0.65–0.90]), 0.76 (95% CI [0.67–0.86]), and 0.68 (95% CI [0.48–0.89]). The mean absolute error (MAE) in the validation cohort was 0.029, indicating a high accuracy. DCA showed a clinical benefit within a threshold probability range of 0.1 to 0.7. **Conclusions:** This study developed a clinically accessible nomogram to predict the risk of MDRO infection in LT recipients, enabling early risk stratification and the real-time assessment of infection risk based on the length of postoperative ICU stay. The model incorporates five easily obtainable clinical parameters (ascites, total bilirubin, albumin, preoperative ICU stay history, and length of ICU stay) and demonstrates strong predictive performance, facilitating the early identification of high-risk patients. Future research should focus on refining the model by incorporating additional clinical factors (e.g., immunosuppressive therapy adherence) and validating its generalizability in multicenter, large-sample cohorts to enhance its clinical utility.

## 1. Introduction

Liver transplantation (LT) is currently the only effective treatment for end-stage liver disease (ESLD); however, postoperative bacterial infections remain one of the main causes of graft dysfunction and death in LT recipients [1,2]. Complex surgical trauma, postoperative immunosuppressive strategies, and various invasive surgical procedures combined with prophylactic antibiotic regimens expose LT recipients to a high risk of infection with multidrug-resistant organisms (MDROs) [3,4]. MDRO infections account for 50% to 70% of bacterial infections post-LT [5], and approximately 31% of recipients experience some degree of MDRO infection within one-year post-surgery [6,7]. The 90-day and one-year mortality rates for LT recipients with MDRO infections are significantly higher than those without MDRO infections [8]. For instance, the 90-day postoperative mortality rate for LT recipients with carbapenem-resistant *Acinetobacter baumannii* (CRAB) infections can reach 46% [9], and the one-year survival rate for LT recipients with carbapenem-resistant *Klebsiella pneumoniae* (CRKP) infections is even lower than 30% [10]. This result is consistent with our previous findings [11]. Compared to ordinary bacterial infections, post-LT MDRO infections often have an insidious onset, rapid progression, and high mortality rate, and most patients die from multiple organ failure caused by severe MDRO infections [12,13,14].

Factors related to MDRO infection after LT permeate the entire perioperative period (preoperative, intraoperative, and postoperative) [5]. Known factors include the following [15,16,17,18]: age, Model for End-Stage Liver Disease (MELD) score, Child–Pugh classification, intraoperative blood loss, the duration of postoperative mechanical ventilation, and the length of stay in the ICU postoperatively. Although many factors related to MDRO infections have been identified, their cumulative effects remain unclear. Additionally, screening potential recipients for MDRO infection is not always straightforward and requires confirmation through a lengthy process of bacterial culture and antibiotic sensitivity testing [19], which can lead to an underestimation of the incidence of MDRO infection and treatment delays [20]. To date, few reliable systems have been able to quantify the risk of MDRO in LT recipients early and accurately [21], despite some studies attempting to do so [22]; however, their sample sizes are limited (n < 200) and have not been validated for internal validity.

Given the serious threat of MDRO infections in LT recipients and the limitations of existing risk assessment tools, it is crucial to develop a tool capable of the early and accurate prediction of MDRO infection risk. A nomogram, a visual tool that integrates multiple factors, can assist clinicians in making clinical decisions. Nomograms are widely used in the LT field [23,24,25] to estimate the incidence or risk of various diseases. For instance, Liu et al. [26] developed a nomogram (validation area under the curve [AUC] 0.81) that predicts the risk of acute kidney injury after LT by inputting clinical parameters (body mass index, hypertension, preoperative creatinine, etc.) with a significant decrease in long-term survival rates in the high-risk group (hazard ratio [HR] 1.92). Xu et al. [27] also developed a nomogram (validation AUC of 0.78) using the postoperative heart rate, creatinine concentration, and blood glucose concentration to predict the risk of sepsis after LT, and compared to the traditional Sequential Organ Failure Assessment (SOFA) score, the nomogram had a higher net benefit rate. This study aimed to develop and validate a nomogram-based prediction model for the early and accurate quantification of MDRO infection risk in LT recipients’ post-surgery, providing clinicians with a practical tool to support timely decision-making and improve patient outcomes.

## 2. Material and Methods

### 2.1. Ethics Approval

The Ethics Review Committee of Beijing Tsinghua Changgung Hospital approved this study (approval number 24206-4-01, approval date 20 March 2024).

### 2.2. Study Design

From January 2018 to December 2023, liver transplant patients admitted to the liver ICU of Beijing Tsinghua Changgung Hospital, affiliated with Tsinghua University, were retrospectively analyzed. The patients were followed up for 90 days postoperatively. The cohort was divided into training and validation sets at a 7:3 ratio. The reason for choosing the 7:3 ratio is as follows. First, this ratio can balance the number of training and validation data in a limited dataset, ensuring that the training set has enough data to learn the features of the model, whereas the validation set can provide enough samples to evaluate the performance of the model. Second, a 7:3 ratio can reduce the risk of overfitting and improve the generalization ability of the model. Compared to other common ratios (such as 8:2 or 9:1), the 7:3 ratio is more suitable for this study because the data volume of multiple drug-resistant bacterial infections after liver transplantation is limited, and it is necessary to retain sufficient validation data to ensure the accuracy of the evaluation. The inclusion criteria were as follows: age ≥ 18 years, absence of combined liver and kidney transplantation, absence of preoperative infection or bacterial colonization, availability of complete clinical data, and survival after 24 h postoperatively. Exclusion criteria included preoperative long-term immunosuppressive treatment (>6 months), transfer to other hospitals during the treatment period, and/or loss to follow-up during the study period. Only the first infection was included in the analysis for patients who experienced multiple MDRO infections during the study period but met the inclusion criteria.

### 2.3. Related Definitions

MDROs are defined as bacteria that simultaneously exhibit resistance to three or more classes of clinically used antibiotics, such as β-lactamase (including penicillins, cephalosporins, and carbapenems), aminoglycosides (such as gentamicin and amikacin), quinolones (such as ciprofloxacin and levofloxacin), glycopeptides (such as vancomycin and teicoplanin), and macrolides (such as azithromycin and clarithromycin) [28]. The definitions of the infections are as follows: (1) Abdominal infection, characterized by abdominal symptoms (e.g., abdominal pain and distension), signs of peritoneal irritation (e.g., tenderness, rebound pain, and muscle tension), abnormal body temperature (>38 °C or <36 °C), blood leukocyte count >10 × 10^9^/L, imaging findings indicating abdominal effusion or abscess formation, and positive culture from abdominal drainage fluid [29]. (2) Pulmonary infection: identified by respiratory symptoms (e.g., cough, expectoration, purulent sputum, or airway secretions, with or without chest pain), abnormal body temperature (>38 °C or <36 °C), imaging changes (e.g., new or progressive pulmonary shadows or interstitial changes on chest X-ray or CT, with or without pleural effusion), and a positive bronchoalveolar lavage fluid culture [11,30].

The testing location was the Clinical Laboratory of Beijing Tsinghua Changgung Hospital. The VITEK-2 compact automated bacterial identification system (bioMérieux, Marcy-l’Étoile, France) and API bacterial identification strips (bioMérieux, Marcy-l’Étoile, France) were used for bacterial identification. Drug susceptibility tests were performed using paper diffusion or dilution methods, and the results were interpreted according to the 2018 Clinical and Laboratory Standards Institute (CLSI) guidelines. The disk diffusion method is simple, inexpensive, and highly flexible, making it suitable for rapid screening and qualitative analysis. Dilution methods (such as the microbroth dilution method) can provide more precise minimum inhibitory concentration (MIC) data, which are suitable for quantitative analysis and research on resistance mechanisms. The combination of these two methods ensures the comprehensiveness and reliability of the results, thus meeting the needs of this study for the assessment of bacterial resistance.

### 2.4. Data Collection and Immunosuppressive Regimen

The clinical data of the recipients were collected through inpatient medical records, medical examination reports, and follow-up evaluations. Data were recorded independently by two individuals to ensure accuracy. If two recorders encounter discrepancies during the data entry process, the following steps were taken to resolve the issue: first, the two recorders would jointly verify the original data (such as medical records and laboratory reports) to confirm whether there was an entry error; second, if the discrepancy could not be resolved through re-verification, a third independent researcher (such as the project leader or a senior clinician) would arbitrate and make a final decision based on the original data and clinical judgment; finally, all discrepancies and their resolution process were documented in detail to ensure the transparency and traceability of data processing. The collected data included patient demographic information (e.g., age, sex, and indications for surgery), preoperative parameters (e.g., antimicrobial use, MELD score, and Child–Pugh score within 30 days before transplantation), and laboratory parameters (e.g., albumin and total bilirubin levels). The MELD and Child–Pugh scores were both calculated based on pre-transplant clinical and laboratory data. Additionally, infection-related data, such as the identification of pathogenic microorganisms and their sites of origin, were reviewed. After liver transplantation, a triple immunosuppressive regimen consisting of tacrolimus (FK506), mycophenolate mofetil (MMF), and glucocorticoids was routinely administered. The target trough level of tacrolimus was 8–12 ng/mL during the first 3 months, 5–10 ng/mL between 3 and 6 months, and 5–8 ng/mL thereafter. Glucocorticoids were tapered and discontinued within 3 months of post-transplantation.

### 2.5. Statistical Analysis and Modeling

Data collection and statistical analyses were performed using R (version 4.2.1; R Foundation for Statistical Computing, Vienna, Austria). Quantitative variables are described as median and interquartile range (IQR) or mean and standard deviation (SD). Count data were compared using the χ^2^ test or Fisher’s exact probability method. Univariate logistic regression analysis was conducted to identify the factors that could be associated with postoperative MDRO infections. Variables with a *p*-value < 0.05 were included in a bidirectional stepwise multivariate regression analysis to establish a logistic model for postoperative MDRO infections. This process was guided by clinical expert opinion to fully account for factor interactions, and potential confounders, such as age, sex, and comorbidities, were adjusted in the multivariate regression models to ensure the reliability of the effect estimates, control for confounders, and improve internal validity. The variable selection process was based on the Akaike Information Criterion (AIC) to minimize model complexity while maximizing the goodness of fit. Effect measures are expressed as odds ratios (ORs) with 95% confidence intervals (CIs).

All statistically significant factors identified through the regression analysis were incorporated into the nomogram prediction model. Nomograms were developed using the “rms” package (version 6.2-0) in R (version 4.2.1). Model performance was evaluated using receiver operating characteristic (ROC) curve analysis, with AUC used to determine the model’s discriminative ability. Calibration was assessed using calibration curves generated by the “rms” package (version 6.2-0), comparing predicted probabilities with actual outcomes using visual plots. While internal validation methods such as ROC and calibration curves were used to assess model performance, external validation or cross-validation was not performed, owing to limitations in the available external datasets. Future studies should validate the model in independent cohorts to confirm its generalizability. Decision curve analysis (DCA) was performed using the “dcurves” package to evaluate the clinical applicability of the nomogram by calculating the net benefit across different threshold probabilities in liver transplant recipients.

## 3. Results

### 3.1. Characteristics of the Study Cohort

A total of 301 LT recipients were enrolled, 210 in the training set and 91 in the validation set. Among them, 56 recipients (18.60%) developed MDRO infections, including 37 (17.62%) in the training cohort and 19 (20.88%) in the validation cohort. The flow diagram of the study design is shown in Figure 1. The cohort comprised 244 males (81.06%) and 57 females (18.94%). The primary indications for liver transplantation were hepatocellular carcinoma (n = 130, 43.19%), hepatitis B virus cirrhosis (n = 79, 26.25%), and alcoholic liver disease (n = 29, 9.63%). The predominant surgical procedure was classic orthotopic liver transplantation (n = 251, 83.39%). Underlying comorbidities included hypertension (n = 36, 11.96%) and diabetes mellitus (n = 49, 16.28%). The additional baseline characteristics of the patients are summarized in Table 1.

### 3.2. Epidemiology and Origin of Pathogenic Bacteria

Among the 301 included recipients, 56 MDRO infections were identified, resulting in an overall infection rate of 18.6%. Gram-negative bacteria (GNB) accounted for most infections (87.5%), with the most common pathogens being *Klebsiella pneumoniae* CRE (46.4%), *Acinetobacter baumannii* CRAB (26.8%), and *Escherichia coli* ESBLs (7.1%). Gram-positive bacteria comprised 12.5% of infections, primarily *Staphylococcus aureus* MRSA (5.4%) (Table 2). The primary sources of infection were the abdomen (55.4%), lungs (19.6%), blood (16.1%), surgical incision sites (5.4%), and infections of unknown origin (3.6%).

### 3.3. Impact of MDRO Infections on Postoperative Outcomes

In the training cohort, recipients in the MDRO group had a significantly longer postoperative ventilator support time (12 h vs. 7 h, *p* = 0.001) and a prolonged length of ICU stay (8.00 days vs. 5.00 days, *p* < 0.001) than those in the non-MDRO group. Additionally, the MDRO group exhibited a higher incidence of acute rejection reactions (56.76% vs. 9.83%, *p* < 0.001) and a markedly increased 90-day all-cause mortality rate (16.22% vs. 1.16%, *p* < 0.001). The details of other variable differences are provided in Table 3.

### 3.4. Multivariate Logistic Regression Model for Predictors of Post-LT MDRO Infection

Guided by clinical expert opinion, univariate logistic regression analysis was conducted to identify relevant parameters. Multivariate bidirectional stepwise logistic regression analysis included factors identified during initial screening. The final regression model incorporated five significant factors: ascites (OR = 3.48, 95% CI [1.33–9.14], *p* = 0.011), total bilirubin (OR = 1.01, 95% CI [1.01–1.01], *p* < 0.001), albumin (OR = 0.85, 95% CI [0.75–0.96], *p* = 0.010), preoperative ICU stay history (OR = 1.09, 95% CI [1.01–1.17], *p* = 0.009), and length of ICU stay (OR = 3.70, 95% CI [1.39–9.84], *p* = 0.019) (Table 4). The discriminative ability of the model was evaluated by ROC curve analysis. The AUC was 0.88 (95% CI [0.81–0.94]) for the training cohort and 0.77 (95% CI [0.65–0.90]) for the validation cohort (Figure 2).

### 3.5. Development of a Nomogram for the Prediction of Post-LT MDRO Infection

A nomogram model (Figure 3) was developed to predict MDRO infection after LT based on multivariate logistic regression analysis. Points were assigned to the four independent predictors according to their regression coefficients, as indicated by the top line of the nomogram labeled “Points”. The total points were calculated to estimate the probability of post-LT MDRO infection.

For example, consider a male patient with ascites, preoperative ICU stay history, 16-day ICU stay, albumin levels of 37 g/L, and total bilirubin levels of 491 μmol/L. Using the nomogram, the total score for this patient was 97, corresponding to approximately an 85% risk of MDRO infection.

The predictive performance of the model was evaluated using the mean absolute error (MAE). In the validation cohort (n = 91), the model demonstrated a low MAE value of 0.029, indicating a high predictive accuracy with a minimal mean difference between the predicted and observed values. The calibration curve of the nomogram (Figure 4A) showed good agreement between the predicted risk of MDRO infection and observed probability. The model decision curve analysis (Figure 4B) demonstrated a threshold probability range of 0.1–0.7, further supporting its clinical utility. These results suggest that the nomogram exhibits acceptable calibration and is reliable for clinical application.

## 4. Discussion

This study identified ascites, total bilirubin, albumin, preoperative ICU stay history, and length of ICU stay as independent factors for MDRO infection after LT. A nomogram model was developed to predict the risk of MDRO infections in LT recipients. Infections occur frequently in LT recipients, with approximately 55% of patients experiencing an infection within 12 months after transplantation [31]. Infection is also the most common cause of death between 30 and 180 days after LT [32]. Unfortunately, an increasing proportion of these infections are caused by MDRO [21]. The MDRO infection rate in this study was 18.6%, which is consistent with the results of previous studies [33,34].

Bacterial intra-abdominal and surgical wound infections remain significant challenges after orthotopic liver transplantation [35]. Our study identified the abdomen as the primary source of infection (55.4%), consistent with the findings of Safdar et al. [36], who reported peritonitis as a common abdominal infection with a less frequent occurrence of liver abscesses. This may be attributed to the complexity of LT procedures and the prolonged nature of dual infections associated with persistent ascites [37]. Furthermore, among LT recipients in the ICU, lower respiratory tract infections, particularly those caused by MDRO, can reach as high as 81% [38]. Among these, the detection rates of *Acinetobacter baumannii* CRAB and *Klebsiella pneumoniae* CRE are increasing [39]. Urinary tract infections are another common source of MDRO infections, with risk factors including urinary tract obstruction and previous antimicrobial treatment [40]. Bloodstream and surgical site infections are also significant factors for MDRO infections in this population, often associated with invasive procedures and prolonged hospitalization [41]. Understanding the diverse sources of MDRO infection is crucial for developing targeted prevention and control strategies. Future studies should explore the epidemiology of these infections and their impact on patient outcomes.

Over the past decade, solid organ transplantation data have steadily increased GNB infections, including an eight-fold increase in MDR GNB [1]. Reports of MDR GNB infections in LT recipients are also increasing, with centers worldwide documenting rates exceeding 50% [7,42]. In this study, 87.5% of the pathogens identified were Gram-negative bacilli, predominantly *Klebsiella pneumoniae* CRE (46.4%) and *Acinetobacter baumannii* CRAB (26.8%). These findings are consistent with those reported by Barchiesi et al. [43], who also identified *Klebsiella pneumoniae* and *Acinetobacter baumannii* as the most common MDR GNB pathogens in LT recipients. However, the prevalence of CRE in our study (46.4%) was higher than that reported in previous studies (e.g., 30% in Barchiesi et al. [43]), which may reflect regional differences in antibiotic resistance patterns or variations in infection control practices.

Among the 301 included recipients, 56 MDRO infections were identified, resulting in an overall infection rate of 18.6%. GNB accounted for most infections (87.5%), with the most common pathogens being *Klebsiella pneumoniae* CRE (46.4%), *Acinetobacter baumannii* CRAB (26.8%), and *Escherichia coli* ESBLs (7.1%). Gram-positive bacteria comprised 12.5% of infections, primarily *Staphylococcus aureus* MRSA (5.4%). The high prevalence of MDR-GNB infections in our study underscores the urgent need for effective infection control measures and tailored antibiotic strategies in LT recipients. LT recipients are particularly prone to CRE infections, which are critical predictors of adverse outcomes [44]. Previous studies [43] have shown that when LT recipients develop CRKP infections, the 1-year survival rate decreases dramatically from 86% to 29% in one study and from 93% to 55% in another [45]. Our findings further highlight the significant impact of MDR-GNB infections on patient outcomes and emphasize the importance of early identification and intervention.

Similarly, in this study, MDRO infections were associated with significantly worse postoperative outcomes, including longer ventilator support time, prolonged ICU stay, and a higher incidence of acute rejection reactions. The markedly higher 90-day all-cause mortality in the MDRO group demonstrates the severity of these infections, consistent with findings from previous studies [11]. The high prevalence of antibiotic resistance and the associated elevated mortality rates highlight the significant therapeutic challenges in managing MDR infections. These findings emphasize the importance of antibiotic stewardship and robust infection-control strategies to mitigate these risks.

This study identified five independent factors associated with MDRO infection after LT: ascites, total bilirubin level, albumin level, preoperative ICU stay history, and length of ICU stay. These findings are consistent with those of previous studies [11,21,34,46]. Unlike the study by Hatice et al. [34], our results demonstrate a significant association between ascites and MDRO infection (OR = 3.48, 95% CI [1.33–9.14]). Ascites is a common and severe complication of ESLD. Patients with ESLD and ascites often require complex treatment and are at an increased risk of complications such as spontaneous bacterial peritonitis (SBP). Among cirrhotic patients with ascites, the incidence of SBP is approximately 27%, with a recurrence rate of 40–70% in those with a history of SBP [47]. Once ascites develops, the one-year mortality rate is approximately 15%, while the five-year mortality rate ranges from 44% to 85% [48]. In this study, 47.51% of the LT recipients presented ascites preoperatively, highlighting the challenges and importance of managing ascites during the perioperative period of LT.

Due to severe liver damage in patients with ESLD, bilirubin metabolism and excretion are significantly impaired, leading to elevated total bilirubin levels. This increase not only reflects liver dysfunction but also the presence of serious complications, such as liver failure or liver encephalopathy. Higher preoperative total bilirubin levels indicate poorer patient status and a higher risk of MDRO infection after complex LT procedures. Hatice et al. [34] similarly reported a significant association between elevated pre-LT bilirubin levels and MDRO infection at 1- and 6-months post-transplantation (*p* = 0.003 and *p* = 0.008, respectively), consistent with our findings.

Although the MELD scores of MDRO-infected recipients were higher in this study, the MELD score was not identified as an independent risk factor. This result is consistent with previous studies [34,49]. However, earlier studies have shown [50] that a MELD score greater than 25 is associated with poorer survival rates for both patients and grafts, especially in the first year after transplantation. Additionally, the MELD score plays a significant role in predicting postoperative complications and mortality. For example, a retrospective analysis based on 135 liver failure LT recipients showed [51] that the initial postoperative MELD score and its derived scores (such as MELD-Na and MELD-Lac) have significant predictive value for early survival rates. Another study indicated [52] that the MELD score is closely related to the risk of postoperative infection, especially when the score is high and patients are more prone to MDRO infections. The MELD score also plays a role in immune regulation after LT, with patients with low MELD scores generally having better immune function and a lower infection risk, whereas patients with high MELD scores may be more susceptible to infections due to severe liver dysfunction and immunosuppression status [53]. Therefore, although the MELD score was not identified as an independent risk factor for MDRO infection in this study, it still plays an important role in assessing the overall postoperative prognosis and infection risk.

This study identified preoperative albumin levels as a protective factor against MDRO infections following LT (OR = 0.85, 95% CI [0.75–0.96]). In the training set, albumin levels in the non-MDRO group were significantly higher than those in the MDRO group (*p* = 0.002). Albumin levels are closely related to the risk of postoperative infection, and maintaining serum albumin levels above 30 g/L during the perioperative period of LT can reduce the incidence of postoperative infections [54]. Studies have reported that preoperative serum albumin levels below 3.0 mg/dL (OR = 0.14) are an independent significant predictor of BSI [55]. For every 0.1 mg/dL increase in albumin levels, the risk of death from BSI after LT decreases by 0.81 times [56], indicating that maintaining higher albumin levels may help reduce the risk of postoperative infections. However, other studies report [57] that there was no significant difference in 1-year mortality rates between the albumin group and the control group, which may be related to the study design, sample size, and observation time. In addition, albumin can alleviate the inflammatory response by inhibiting the production of proinflammatory cytokines and regulating the function of immune cells [58]. Albumin also plays a key role in maintaining endothelial stability and regulating immune responses, and it can stabilize endothelial cells, reduce endothelial damage induced by oxidative stress, protect cells from damage by binding to reactive oxygen species and reactive nitrogen species [59], and enhance immune defense capabilities by regulating the function of T cells and B cells [60].

Preoperative ICU stay history is a significant risk factor for MDRO infection [11]. In this study, twenty patients in the training set had a history of preoperative ICU stay, of which eight (40%) developed MDRO infection. In the validation set, seven patients had a history of preoperative ICU stay, of which two (28.6%) developed MDRO infection. These data further support the significant association between a history of preoperative ICU stay and MDRO infection. Previous studies have shown that preoperative ICU stay history is an independent risk factor for BSI caused by multidrug-resistant *Pseudomonas aeruginosa* (OR = 2.04, 95% CI [1.15~3.63]) [61]. Preoperative ICU stay history may act in conjunction with other high-risk factors (such as invasive procedures and antimicrobial use) to further increase the risk of MDRO infection [62]. Consistent with the data from this study, previous research has reported a correlation between postoperative ICU stay and MDRO infection after LT [19,63,64], with the risk of postoperative MDRO infection increasing with the length of ICU stay. We once again emphasize the need for focused attention and effective prevention and control of infection status for LT recipients with longer postoperative ICU stays.

In recent years, transplant experts have committed to identifying perioperative risk factors for MDRO infections after LT. However, there remains a lack of well-performing predictive models. The present study fills this gap by constructing a nomogram model. This model demonstrated strong predictive performance and good discrimination and calibration ability in both the training and validation sets. Specifically, the AUC value of the validation set (0.77) was slightly lower than that of the training set (0.88), which may be attributed to several reasons. First, there may be differences in patient characteristics between the training and validation sets (such as preoperative ICU stay history), and a higher proportion of high-risk patients in the validation set may affect the generalizability of the model [65]. Second, the smaller sample size of the validation set may have led to fluctuations in AUC values. In addition, the different MDRO infection rates between the validation and training sets may further affect the predictive performance of the model [66]. Despite this, the model’s calibration curve shows a good overlap between the predicted and actual values, indicating that the model still has high accuracy on the validation set [67]. Future studies can improve the generalization ability of the model by expanding the sample size of the validation set, optimizing the data balance, and further adjusting the model parameters.

DCA was used to evaluate the clinical utility of the predictive model by calculating net clinical benefit. The findings suggest that nomogram-based interventions provide meaningful outcomes when the probability threshold is 0.1 and 0.7. A key strength of this model is its inclusion of clinically accessible and quantifiable parameters, enabling healthcare professionals to quickly calculate the risk of MDRO infection in potential recipients and monitor risk changes over time. This facilitates the early identification of high-risk patients, allowing the development of individualized perioperative strategies to prevent the emergence and spread of MDR strains and to mitigate their impact.

This study has several limitations. First, this was a single-center retrospective study with a relatively limited number of LT recipients. Consequently, the external validation of the model was not performed. External validation is crucial for assessing the generalizability of predictive models across different populations and clinical settings [68]. The lack of external validation may limit the confidence in the model’s applicability to other cohorts. Future prospective multicenter studies are needed to validate these findings and improve the generalizability of our observations. Additionally, spatial validation (e.g., testing the model in different geographic or clinical settings) should be considered to further assess the model’s robustness and clinical utility. Second, this study did not analyze strict adherence to immunosuppressive drug therapy or medical follow-up. Immunosuppressive therapy adherence is a critical factor influencing post-transplant outcomes, including the risk of MDRO infection [69]. Nonadherence may lead to suboptimal immunosuppression, increasing the risk of rejection and infection. Future studies should incorporate detailed assessments of immunosuppressive therapy adherence using accurate monitoring methods (e.g., drug concentration measurements or patient self-reports) to provide a more comprehensive understanding of MDRO risk and post-transplant outcomes [70].

## 5. Conclusions

MDRO infection is closely associated with adverse outcomes after LT. This study, based on patient clinical data analysis, developed an easy-to-use nomogram model that provides a valuable tool for the prevention and control of MDRO infection. This model integrates multiple risk factors (such as preoperative ICU stay history and albumin levels) to help clinicians identify high-risk patients early, optimize antibiotic use strategies, and take targeted preventive measures. This approach aims to reduce the incidence of complications related to MDRO infections and improve patient outcomes. Future research should further validate and optimize the model through multicenter, large-sample cohorts, explore other potential risk factors (such as compliance with immunosuppressive therapy and postoperative care quality), and evaluate its generalizability in different clinical settings. Additionally, incorporating machine learning or deep learning technologies may further enhance the predictive accuracy and stability of the model. Ultimately, the effectiveness and stability of the model should be reflected in its improvement of clinical outcomes (such as infection and mortality rates), providing strong support for the personalized management of LT recipients.

## Figures and Tables

**Figure 1 bioengineering-12-00417-f001:**
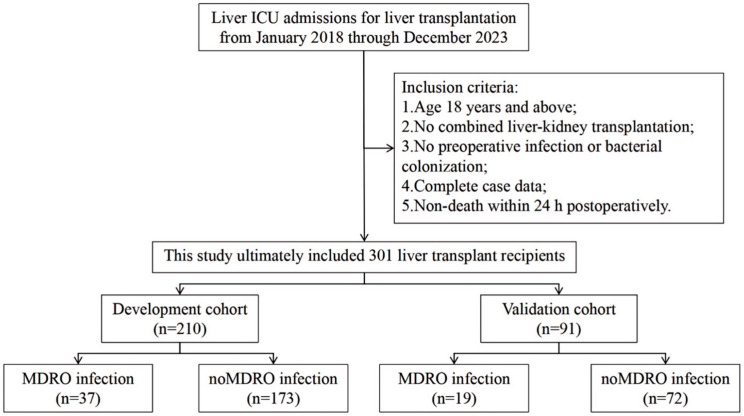
The study design flow diagram.

**Figure 2 bioengineering-12-00417-f002:**
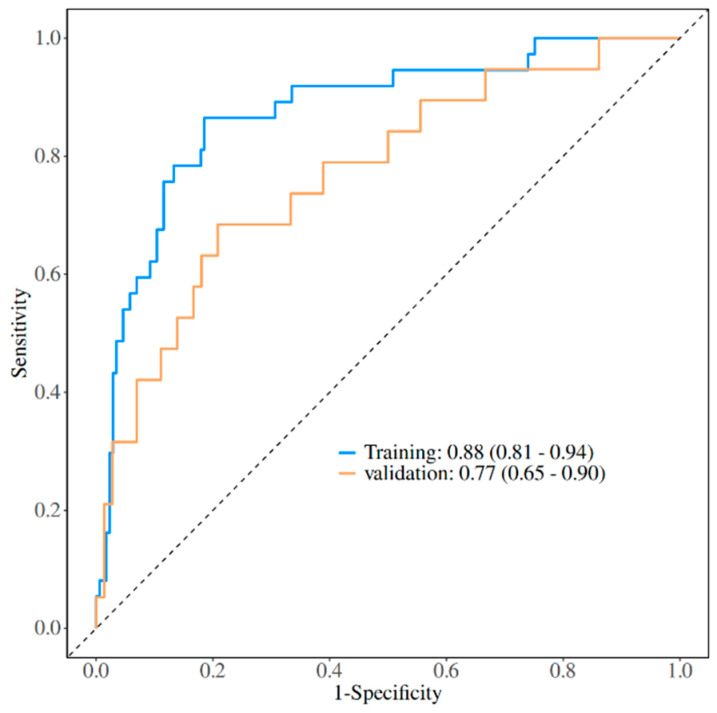
The discriminative ability of the model was evaluated using ROC curve analysis. The AUC is 0.88 (95% CI [0.81–0.94]) for the training cohort and 0.77 (95% CI [0.65–0.90]) for the validation cohort.

**Figure 3 bioengineering-12-00417-f003:**
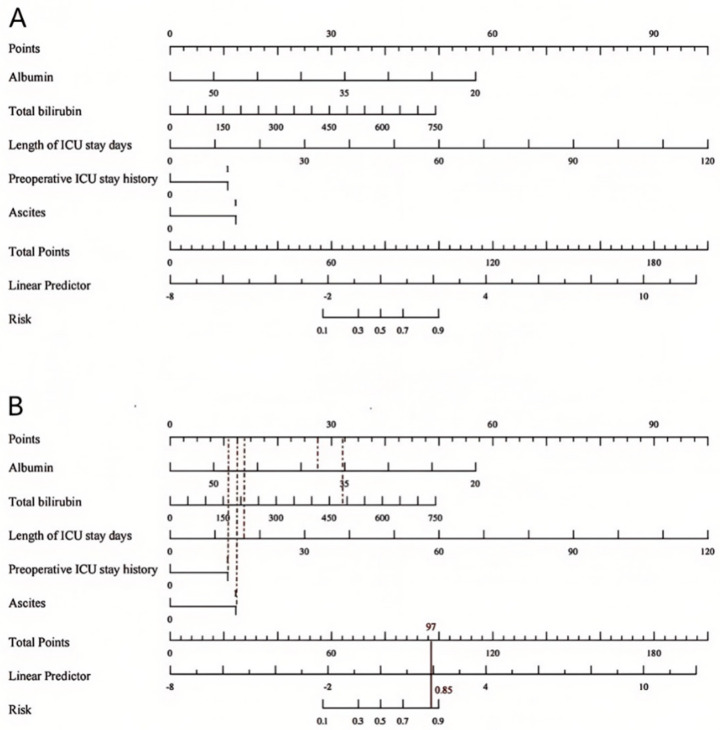
(**A**) For nomogram model. (**B**) A reference example for how to use this model.

**Figure 4 bioengineering-12-00417-f004:**
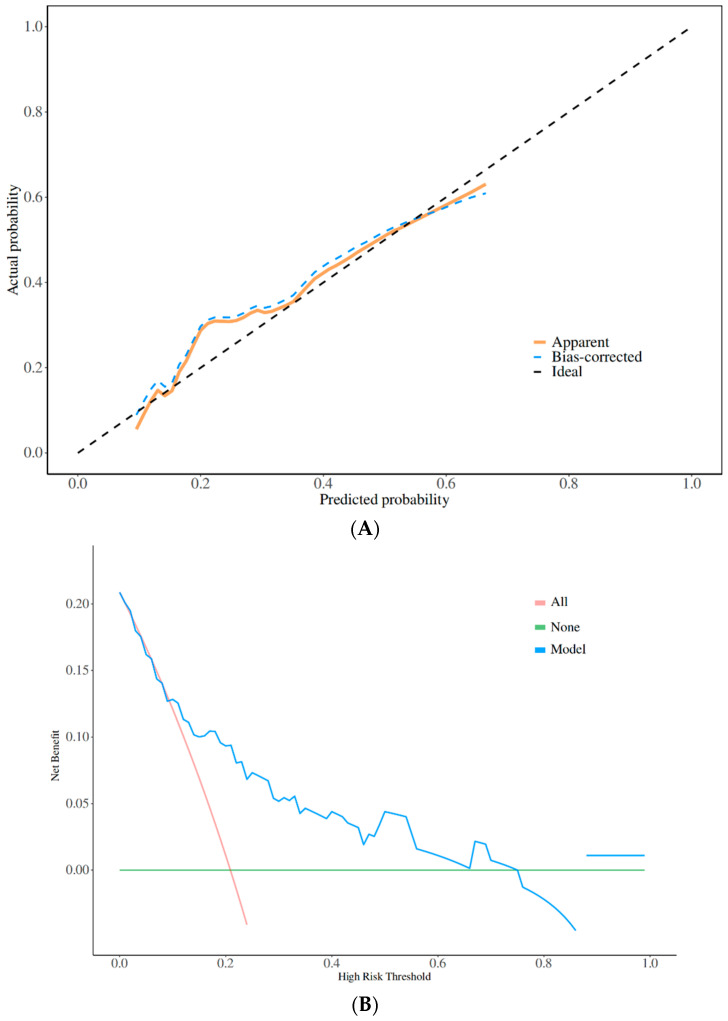
The calibration curve of the nomogram (**A**) shows good agreement between the predicted risk of MDRO infection and the observed probability. The model decision curve analysis (**B**) demonstrates a threshold probability range of 0.1–0.7, further supporting its clinical utility.

**Table 1 bioengineering-12-00417-t001:** Characteristics of study population.

Characteristic	Total	Training Cohort	Validtion Cohort
	(n = 301)	(n = 210)	(n = 91)
Age (years)	53.0 (46.0, 60.0)	53.0 (46.0, 59.8)	52.0 (46.0, 60.5)
Sex			
Male	244 (81.1%)	79 (86.8%)	165 (78.6%)
Female	57 (18.9%)	12 (13.2%)	45 (21.4%)
BMI (kg/m^2^)	22.0 (20.0, 24.0)	22.0 (20.0, 24.0)	22.0 (21.0, 24.0)
Primary disease for LT			
Hepatocellular carcinoma	130 (43.2%)	91 (43.3%)	39 (42.9%)
Hepatitis B virus cirrhosis	79 (26.3%)	56 (26.7%)	23 (25.3%)
Alcoholic liver cirrhosis	29 (9.6%)	18 (8.6%)	11 (12.1%)
Acute liver failure	18 (6%)	13 (6.2%)	5 (5.5%)
Primary biliary cirrhosis	13 (4.3%)	12 (5.7%)	1 (1.1%)
Autoimmune liver disease	10 (3.3%)	6 (2.9%)	4 (4.4%)
Cholangiocarcinoma	8 (2.7%)	4 (1.9%)	4 (4.4%)
Hepatitis C virus cirrhosis	5 (1.7%)	4 (1.9%)	1 (1.1%)
Others	9 (3%)	6 (2.9%)	3 (3.3%)
Comorbidities			
Diabetes	49 (16.3%)	37 (17.6%)	12 (13.2%)
Hypertension	36 (12%)	23 (11%)	13 (14.3%)
Hypertension and diabetes	15 (5%)	9 (4.3%)	6 (6.6%)
Operation			
Classic orthotopic liver transplantation	251 (83.4%)	178 (84.8%)	73 (80.2%)
Piggyback liver transplantation	46 (15.3%)	29 (13.8%)	17 (18.7%)
Split liver transplantation	4 (1.3%)	3 (1.4%)	1 (1.1%)
Ascites	143 (47.5%)	96 (45.7%)	47 (51.7%)
Biliary intestinal anastomosis	33 (11%)	19 (9.1%)	14 (15.4%)
MELD score ≥25	34 (11.3%)	23 (11%)	11 (12.1%)
Child-Pugh score C	32 (10.6%)	23 (11%)	9 (9.9%)
Hepatic encephalopathy	28 (9.3%)	18 (8.6%)	10 (11%)
Antibacterial drug use within 30 days before LT	25 (8.3%)	19 (9.1%)	6 (6.6%)
Preoperative ICU stay history	27 (9%)	20 (9.5%)	7 (7.7%)
Surgical time (h)	8.0 (7.0, 9.0)	8.0 (7.0, 9.0)	8.0 (8.0, 10.0)
Intraoperative blood loss (ml)	400.0 (200.0, 600.0)	400.0 (200.0, 600.0)	400.0 (200.0, 600.0)
Total bilirubin (μmol/L)	137.0 (100.0, 210.0)	132.0 (99.3, 212.3)	139.0 (100.0, 208.5)
Albumin (g/L)	35.0 (32.0, 37.0)	35.0 (32.0, 37.0)	35.0 (31.0, 37.0)
MDRO	56 (18.6%)	37 (17.6%)	19 (20.9%)
WBC (*10^9^/L)	5.8 (4.6, 8.6)	5.9 (4.6, 8.6)	5.6 (4.5, 8.2%)
Transplantation or open laparotomy again	12 (4%)	10 (4.8%)	2 (2.2%)
CRRT	12 (4%)	7 (3.3%)	5 (5.5%)
Biliary complications	20 (6.6%)	14 (6.7%)	6 (6.6%)
Intestinal complications	17 (5.7%)	15 (7.1%)	2 (2.2%)
Ventilator support time (h)	8.0 (6.0, 10.0)	8.0 (6.0, 10.0)	8.0 (6.0, 10.0)
Length of ICU stay days	6.0 (4.0, 8.0)	5.5 (4.0, 8.0)	6.0 (4.0, 8.0)
Acute rejection reaction	48 (16%)	38 (18.1%)	10 (11%)
Total length of hospital stay (d)	16.0 (12.0, 22.0)	16.0 (12.0, 23.0)	16.0 (11.0, 20.5)
28 d all-cause mortality	4 (1.3%)	3 (1.4%)	1 (1.1%)
90 d all-cause mortality	9 (3%)	8 (3.8%)	1 (1.1%)

BMI: body mass index; MDRO: multi-drug resistant organisms; WBC: white blood cell; CRRT: continuous renal replacement therapy.

**Table 2 bioengineering-12-00417-t002:** Between-group comparison of parameters in training cohort liver transplant recipients.

Variables	Total	No MDRO	MDRO	*p*-Value
	(n = 210)	(n = 173)	(n = 37)	
Age (years)	53.0 (46.0, 59.8)	53.0 (46.0, 59.0)	56.0 (42.0, 62.0)	0.720
Sex				0.975
Male	165 (78.6%)	136 (78.6%)	29 (78.4%)	
Female	45 (21.4%)	37 (21.4%)	8 (21.6%)	
BMI (kg/m^2^)	22.0 (20.0, 24.0)	22.0 (20.0, 24.0)	22.0 (21.0, 24.0)	0.337
Primary disease for LT				
Hepatocellular carcinoma	91 (43.3%)	76 (43.9%)	15 (40.5%)	0.129
Hepatitis B virus cirrhosis	56 (26.7%)	50 (28.9%)	6 (16.2%)	
Alcoholic liver cirrhosis	18 (8.6%)	14 (8.1%)	4 (10.8%)	
Acute liver failure	13 (6.2%)	8 (4.6%)	5 (13.5%)	
Primary biliary cirrhosis	12 (5.7%)	11 (6.4%)	1 (2.7%)	
Autoimmune liver disease	6 (2.9%)	4 (2.3%)	2 (5.4%)	
Cholangiocarcinoma	4 (1.9%)	2 (1.2%)	2 (5.4%)	
Hepatitis C virus cirrhosis	4 (1.9%)	3 (1.7%)	1 (2.7%)	
Others	6 (2.9%)	5 (2.9%)	1 (2.7%)	
Comorbidities				0.166
Diabetes	37 (17.6%)	27 (15.6%)	10 (27%)	
Hypertension	23 (11%)	20 (11.6%)	3 (8.1%)	
Hypertension and diabetes	9 (4.3%)	6 (3.5%)	3 (8.1%)	
Operation				0.049
Classic orthotopic liver transplantation	178 (84.8%)	150 (86.7%)	28 (75.7%)	
Piggyback liver transplantation	29 (13.8%)	22 (12.7%)	7 (18.9%)	
Split liver transplantation	3 (1.4%)	1 (0.6%)	2 (5.4%)	
Ascites	96 (45.7%)	68 (39.3%)	28 (75.7%)	<0.001
Biliary intestinal anastomosis	19 (9.1%)	13 (7.5%)	6 (16.2%)	0.174
MELD score ≥25	23 (11%)	16 (9.3%)	7 (18.9%)	0.156
Child-Pugh score C	23 (11%)	15 (8.7%)	8 (21.6%)	0.046
Hepatic encephalopathy	18 (8.6%)	14 (8.1%)	4 (10.8%)	0.832
Antibacterial drug use within 30 days before LT	19 (9.1%)	13 (7.5%)	6 (16.2%)	0.174
Preoperative ICU stay history	20 (9.5%)	12 (6.9%)	8 (21.6%)	0.014
Surgical time (h)	8.0 (7.0, 9.0)	8.0 (7.0, 9.0)	9.0 (8.0, 12.0)	<0.001
Intraoperative blood loss (ml)	400.0 (200.0, 600.0)	400.0 (200.0, 500.0)	600.0 (300.0, 800.0)	<0.001
Total bilirubin (μmol/L)	132.0 (99.3, 212.3)	116.0 (95.0, 200.0)	217.0 (187.0, 379.0)	<0.001
Albumin (g/L)	35.0 (32.0, 37.0)	35.0 (32.0, 38.0)	33.0 (31.0, 35.0)	0.002
WBC (*10^9^/L)	5.9 (4.6, 8.6)	5.9 (4.6, 8.6)	5.9 (5.0, 8.5)	0.639
Transplantation or open laparotomy again	10 (4.8%)	6 (3.5%)	4 (10.8%)	0.139
CRRT	7 (3.3%)	5 (2.9%)	2 (5.4%)	0.788
Biliary complications	14 (6.7%)	8 (4.6%)	6 (16.2%)	0.028
Intestinal complications	15 (7.1%)	11 (6.4%)	4 (10.8%)	0.547
Ventilator support time (h)	8.0 (6.0, 10.0)	7.0 (6.0, 9.0)	12.0 (6.0, 23.0)	0.001
Length of ICU stay days	5.5 (4.0, 8.0)	5.0 (4.0, 7.0)	8.0 (6.0, 15.0)	<0.001
Acute rejection reaction	38 (18.1%)	17 (9.8%)	21 (56.8%)	<0.001
Total length of hospital stay (d)	16.0 (12.0, 23.0)	15.0 (12.0, 20.0)	38.0 (21.0, 43.0)	<0.001
28 d all-cause mortality	3 (1.4%)	2 (1.2%)	1 (2.7%)	0.443
90 d all-cause mortality	8 (3.8%)	2 (1.2%)	6 (16.2%)	<0.001

BMI: body mass index; WBC: white blood cell; CRRT: continuous renal replacement therapy.

**Table 3 bioengineering-12-00417-t003:** Distribution of resistance rates of pathogenic bacteria.

Pathogen	The Number of Pathogen Detections (56)	Percentage (%)
Drug-resistant gram-negative bacteria	49	0.875
Klebsiella pneumoniae CRE	26	0.464
Acinetobacter baumannii CRAB	15	0.268
Escherichia coli ESBLs	4	0.071
Enterobacter cloacae CRE	2	0.036
Pseudomonas aeruginosa CRPA	2	0.036
Drug-resistant gram-positive bacteria	7	0.125
Staphylococcus aureus MRSA	3	0.054
Staphylococcus epidermidis MRS	1	0.018
Staphylococcus caprae MRS	1	0.018
Staphylococcus hominis MRS	1	0.018
Enterococcus faecium VRE	1	0.018

CRAB, carbapenem-resistant Acinetobacter; CRE, carbapenem resistant Enterobacteriaceae; ESBL, extended-spectrum beta lactamase; VRE, vancomycin-resistant enterococcus; MRS, methicillin-resistant staphylococcus.

**Table 4 bioengineering-12-00417-t004:** Multivariate logistic regression model for the predictors of post-LT MDRO infection.

	Univariate Analysis			Multivariate Analysis		
	β	OR (95%CI)	*p*-Value	β	OR (95%CI)	*p*-Value
Ascites	1.57	4.80 (2.14~10.81)	<0.001	1.25	3.48 (1.33~9.14)	0.011
Child-Pugh score C	1.07	2.91 (1.13~7.48)	0.027			
Preoperative ICU stay history	1.31	3.70 (1.39~9.84)	0.009			
Total bilirubin (μmol/L)	0.01	1.01 (1.01~1.01)	<0.001	0.01	1.01 (1.01~1.01)	<0.001
Albumin (g/L)	−0.17	0.84 (0.76~0.93)	<0.001	−0.17	0.85 (0.75~0.96)	0.010
Biliary complications	1.38	3.99 (1.29~12.31)	0.016			
Length of ICU stay days	0.14	1.15 (1.07~1.23)	<0.001	0.09	1.09 (1.01~1.17)	0.019

## Data Availability

The raw data supporting the conclusions of this article will be made available by the authors on request.

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
