# Peer review of "Epidemiology and Risk Prediction Model of Multidrug-Resistant Organism Infections After Liver Transplant Recipients: A Single-Center Cohort Study"

_bioengineering, 2025, doi:10.3390/bioengineering12040417_

Round 1
Reviewer 1 Report
Comments and Suggestions for Authors
Dear Authors,
Thank you for the opportunity to review your manuscript. I appreciate the effort and thoroughness you have put into this study, and I believe the findings have important implications for improving the management of MDRO infections in liver transplant recipients. I hope the comments provided will assist in further refining and strengthening your work.
I look forward to seeing the revised version of your manuscript and wish you all the best in your continued research.
Title & Abstract:
- Title: The title could be more precise in reflecting both the risk prediction model and the epidemiology of MDRO infections in liver transplant recipients. A more comprehensive title would provide readers with a clearer understanding of the study’s purpose.
- Objective: It would be helpful to include a brief explanation of the clinical significance of predicting MDRO infections in LT recipients. This will help emphasize why developing a prediction model is crucial for improving patient outcomes.
- Cohort Size and Characteristics: The study cohort sizes in the training and validation sets should be explained further. It would be beneficial to provide the rationale for the different sizes (7:3 ratio) and how this might influence the results. Additionally, a discussion of potential biases that could arise from these cohort sizes would be valuable.
- Model Evaluation: While the statistical methods are well described, adding more details regarding the model's sensitivity, specificity, and how it compares to other similar prediction models would improve the clarity and rigor of the performance evaluation.
- Clinical Application: It would be useful to discuss how the nomogram can be implemented in real-world clinical practice. This would help clinicians understand its practical value in decision-making.
- Future Research: The paper would benefit from suggestions regarding potential improvements to the prediction model or further studies aimed at refining its predictive capabilities. This could include incorporating additional clinical factors or utilizing different populations.
Introduction and Study Aim:
- Clarify the Challenge of MDRO Infections: The introduction provides valuable information, but a clearer explanation of the challenges posed by MDRO infections specifically in liver transplant recipients would help readers appreciate the study’s relevance.
- Smooth Transition to Study Aim: Consider a smoother transition from the background information on MDRO infections to the study’s aim. This would help the reader understand the context more logically.
- Organize References More Cohesively: The references should be organized in a more cohesive manner. Grouping similar studies or discussing trends over time could enhance the flow and depth of the introduction.
- Enhance Nomogram Application: While the nomogram's development is well described, expanding on how it can be practically used by clinicians, and how it might impact patient management, would strengthen this section.
- Reinforce Study Impact: More emphasis on how this study could change current clinical practices in liver transplantation or MDRO infection management would be beneficial.
Material and Methods:
- Inclusion Criteria: The exclusion of patients with preoperative infections or bacterial colonization may be overly restrictive. A more detailed justification for excluding these patients would help clarify why this criterion was chosen, especially given the potential prevalence of these factors in real-world clinical practice.
- Follow-up Period: Please clarify why a 90-day follow-up period was chosen. Discuss whether this timeframe sufficiently captures the key outcomes of MDRO infections and whether a longer follow-up period might have offered additional insights.
- Cohort Split: The rationale for the 7:3 training to validation cohort ratio should be included. While this is a common approach, explaining why this ratio was chosen specifically for this study, and how it might influence the model's generalizability, would strengthen the methodology section.
- Exclusion Criteria: The term "prolonged immunosuppressive therapy" should be more clearly defined to avoid ambiguity. A specific threshold or definition (e.g., duration or dosage) would provide more clarity.
- MDR Definition: It would be helpful to include specific examples of antimicrobial categories (e.g., beta-lactams, aminoglycosides, etc.) when defining multidrug resistance (MDR). This will aid in clarifying the specific resistance patterns under consideration.
- Celiac Infection Clarification: The term "Celiac infection" appears to be a typo or incorrect. Please verify and correct it to ensure accuracy in the description of the infections.
- Leukocyte Count Criteria: The criteria for an elevated peripheral blood leukocyte count could benefit from a clearer range. Providing specific thresholds (e.g., >10×10⁹/L) would make the definition more precise.
- Antibiotic Susceptibility Testing Methods: It would be helpful to mention the strengths of the methods used (e.g., disc diffusion vs. dilution) for antibiotic susceptibility testing to justify their appropriateness for the study.
- Data Collection Process: The description of data collection is generally clear. However, it would be useful to explain how discrepancies between the two independent recorders were resolved to ensure data reliability.
- Preoperative Parameters: Clarify whether MELD and Child-Pugh scores were calculated using pre-transplant values or baseline scores for the patients. This distinction is important for reproducibility.
- Immunosuppressive Regimen: For clarity, it would be useful to mention the target levels for tacrolimus and the duration of glucocorticoid treatment. This additional information would help replicate the immunosuppressive regimen used in this study.
- Statistical Software: In addition to the R version mentioned, specifying the R package versions used for the statistical methods (e.g., "rms" and "dcurves") would help with reproducibility.
- Variable Selection Process: The bidirectional stepwise regression method is well described. However, it would be beneficial to clarify the specific criteria for variable inclusion/exclusion (e.g., AIC or BIC criteria).
- Effect Measures: It would be helpful to mention whether any adjustments were made for potential confounders in the regression models. This ensures that the model’s estimates are reliable.
- DCA in Deceased Organ Donors: The mention of DCA in deceased organ donors seems out of context since this study focuses on liver transplant recipients. This section should be revised to either reflect the correct context or clarify if a specific subgroup was analyzed.
- Model Evaluation: While the ROC curve and calibration methods are well explained, it would be useful to mention if cross-validation or external validation was used to assess the model’s generalizability to other populations.
Discussion:
- Infection Sources: Expand on other potential sources of MDRO infections beyond abdominal infections (e.g., respiratory, urinary tract). This would provide a more comprehensive view of the infection sources.
- MDR GNB Infections: It would strengthen the discussion to include data from this study on the prevalence of MDR GNB pathogens and compare it with previous literature.
- Albumin's Role: Please include a discussion on how albumin affects immune modulation and resistance to infections. This factor could have a significant impact on post-transplant infection risk.
- Preoperative ICU Stay: Add specific numbers or percentages of patients with a preoperative ICU stay who developed MDRO infections. This would provide more granular insights into the impact of ICU stays on infection risk.
- MELD Scores: Clarify the role of MELD scores in post-transplant prognosis. Specifically, explain how MELD scores correlate with MDRO infection risk or other post-transplant outcomes.
- Model Evaluation (AUC): Explain the slight difference in AUC between the training and validation cohorts (0.88 vs. 0.77). Discuss whether this could be due to differences in cohort characteristics or MDRO infection rates.
- Limitations: Please address the impact of lack of external validation on the model’s generalizability. External validation would help improve confidence in the model’s applicability to different populations.
- Immunosuppressive Therapy: Finally, future studies should consider examining the adherence to immunosuppressive therapy, as it is likely to influence infection risk and may provide a more complete understanding of MDRO risk and post-transplant outcomes.
Conclusions:
- Clarity and Emphasis on Model's Impact: The conclusion could be more specific about how the nomogram contributes to clinical practice. Instead of simply stating that it is an "easy-to-use" tool, it would be helpful to highlight how it might influence clinical decision-making or patient management specifically. For example, how could the nomogram help clinicians identify high-risk patients earlier or adjust treatment strategies?
- Future Studies: While the suggestion for large-sample multicenter cohorts is appropriate, it would be beneficial to mention what specific aspects of the model need further refinement or validation. For example, are there any particular variables that need further exploration, or is the focus on validating the model's generalizability and performance across different settings?
- Model Efficacy and Stability: The reference to assessing efficacy and stability in clinical practice is a good direction, but adding more details on what these terms mean in the context of the model's application would improve clarity. Does "efficacy" refer to its accuracy, or its impact on clinical outcomes like infection rates or mortality?
Comments on the Quality of English Language
Please refer to the comments sent to the authors for parts that require improvement in English.
Author Response
Title & Abstract:
Comment 1: Title: The title could be more precise in reflecting both the risk prediction model and the epidemiology of MDRO infections in liver transplant recipients. A more comprehensive title would provide readers with a clearer understanding of the study’s purpose.
Response 1: Thank you for pointing this out. We agree with this comment. We have revised the title. This revised title more clearly highlights the two main components of our study: (1) the epidemiological characteristics of MDRO infections in this population, and (2) the establishment of a risk prediction model. See lines 2-3 on the first page of the revision and marked in red.
Comment 2: Objective: It would be helpful to include a brief explanation of the clinical significance of predicting MDRO infections in LT recipients. This will help emphasize why developing a prediction model is crucial for improving patient outcomes.
Response 2: Thank you for pointing this out. We agree with this comment. We have modified the objective section of the abstract to better clarify the necessity of developing a prediction model for MDRO infections in liver transplant recipients. See lines 16-20 on the first page of the revision and marked in red.
Comment 3: Cohort Size and Characteristics: The study cohort sizes in the training and validation sets should be explained further. It would be beneficial to provide the rationale for the different sizes (7:3 ratio) and how this might influence the results. Additionally, a discussion of potential biases that could arise from these cohort sizes would be valuable.
Response 3: Thank you for pointing this out. We agree with this comment. We further explained the research cohort sizes of the training set and the validation set. See lines 21-25 on the first page of the revision and marked in red.
Comment 4: Model Evaluation: While the statistical methods are well described, adding more details regarding the model's sensitivity, specificity, and how it compares to other similar prediction models would improve the clarity and rigor of the performance evaluation.
Response 4: Thank you for pointing this out. We agree with this comment. We have supplemented the details of the model for training and validation sets at the relevant positions in the revised draft based on this suggestion. See lines 36-40 on page 1-2 of the revision and marked in red.
Comment 5: Clinical Application: It would be useful to discuss how the nomogram can be implemented in real-world clinical practice. This would help clinicians understand its practical value in decision-making.
Response 5: Thank you for pointing this out. We agree with this comment. We have further discussed the practical value of nomograms in the real world at the relevant positions in the revised draft. See lines 44-50 on page 2 of the revision and marked in red.
Comment 6: Future Research: The paper would benefit from suggestions regarding potential improvements to the prediction model or further studies aimed at refining its predictive capabilities. This could include incorporating additional clinical factors or utilizing different populations.
Response 6: Thank you for pointing this out. We appreciate the reviewer’s valuable suggestion. In the revised manuscript, we have expanded the discussion on future research. Specifically, we propose refining the model by incorporating additional clinical factors, such as immunosuppressive therapy adherence, and validating its generalizability in multicenter, large-sample cohorts. These improvements aim to enhance the model’s predictive capabilities and clinical applicability. Please refer to the revision on page 2, lines 50-54 marked in red.
Introduction and Study Aim:
Comment 1: Clarify the Challenge of MDRO Infections: The introduction provides valuable information, but a clearer explanation of the challenges posed by MDRO infections specifically in liver transplant recipients would help readers appreciate the study’s relevance.
Response 1: Thank you for pointing this out. We have made modifications to the original text based on your suggestions. Please refer to the revision on page 3, lines 80-99, marked in red.
Comment 2: Smooth Transition to Study Aim: Consider a smoother transition from the background information on MDRO infections to the study’s aim. This would help the reader understand the context more logically.
Response 2: Thank you for pointing this out. We have made modifications to the original text based on your suggestions. Please refer to the revision on page 3, lines 100-114, marked in red.
Comment 3: Organize References More Cohesively: The references should be organized in a more cohesive manner. Grouping similar studies or discussing trends over time could enhance the flow and depth of the introduction.
Response 3: Thank you for pointing this out. We have made modifications to the original text based on your suggestions. Please refer to the revision on page 3-4, lines 115-133, marked in red.
Comment 4: Enhance Nomogram Application: While the nomogram's development is well described, expanding on how it can be practically used by clinicians, and how it might impact patient management, would strengthen this section.
Response 4: Thank you for pointing this out. We have made modifications to the original text based on your suggestions. Please refer to the revision on page 3-4, lines 119-129, marked in red.
Comment 5: Reinforce Study Impact: More emphasis on how this study could change current clinical practices in liver transplantation or MDRO infection management would be beneficial.
Response 5: Thank you for pointing this out. We have made modifications to the original text based on your suggestions. Please refer to the revision on page 4, lines 130-133, marked in red.
Material and Methods:
Comment 1: Inclusion Criteria: The exclusion of patients with preoperative infections or bacterial colonization may be overly restrictive. A more detailed justification for excluding these patients would help clarify why this criterion was chosen, especially given the potential prevalence of these factors in real-world clinical practice.
Response 1: Thank you for pointing this out. Regarding the concern that "excluding patients with preoperative infection or bacterial colonization may be overly strict," we fully understand your worries. However, the reason for choosing this criterion at the outset of this study is based on the following considerations: (1) Preoperative infection or bacterial colonization is an important risk factor for postoperative infection [1], excluding these patients helps to analyze other independent risk factors for postoperative infection more clearly; (2) This criterion can minimize the impact of confounding factors on the research results, thereby improving the internal validity of the study; (3) Although this criterion may limit the universality of the sample, it may ensure the rigor and reliability of the research results. Our inclusion criteria are not designed in isolation but are based on relevant literature and clinical practice. For example, studies by Chen, Wang, and others [2, 3] also adopted similar standards to exclude the interference of preoperative infections on postoperative outcomes. This indicates that although excluding preoperative infection patients may limit the representativeness of the sample, it helps to more clearly identify other risk factors for postoperative infection. As you have stated, preoperative infection or bacterial colonization may be relatively common among transplant patients. We have clearly pointed out this limitation in the discussion section, and future studies can further validate our findings under more relaxed inclusion criteria. For example, a standard of 'no clinical or laboratory evidence of infection for at least three consecutive days prior to transplantation' could be adopted to better reflect the actual clinical situation.
Relevant references:
[1]Freire MP, Oshiro IC, Pierrotti LC, et al. Carbapenem-Resistant Enterobacteriaceae Acquired Before Liver Transplantation: Impact on Recipient Outcomes. Transplantation. 2017;101(4):811-820. doi:10.1097/TP.0000000000001620.
[2]Chen Y, Wang WL, Zhang W, et al. Risk Factors and Outcomes of Carbapenem-Resistant Enterobacteriaceae Infection After Liver Transplantation: A Retrospective Study in a Chinese Population. Infect Drug Resist. 2020;13:4039-4045. Published 2020 Nov 10. doi:10.2147/IDR.S278084.
[3]Wang D, Zhou S, Wang Y, et al. Clinical features and risk factors of multi-drug resistant bacteria infection after liver transplantation. Chinese Journal of Hepatobiliary Surgery, 2023, 29(9): 646-650. doi:10.3760/cma.j.cn113884-20230227-00055.
Comment 2: Follow-up Period: Please clarify why a 90-day follow-up period was chosen. Discuss whether this timeframe sufficiently captures the key outcomes of MDRO infections and whether a longer follow-up period might have offered additional insights.
Response 2: Thank you for pointing this out. Regarding the selection of the follow-up period, after thorough discussion, we believe that a 90-day (i.e., 3-month) follow-up period is reasonable. The reason is that multiple studies have shown that the incidence of MDRO infections is higher in the early post-liver transplantation period, especially within the first three months. For instance, a multicenter study in Spain [1] showed that among 960 liver transplant patients, the median time to the first MDRO infection was 11 days (IQR 4-22); furthermore, a meta-analysis [2] indicated that the pooled incidence of MDRO infections post-liver transplantation was 18%, with most infections occurring within the first three months post-surgery. In our previous study [3], the average duration of the first episode of MDRO bloodstream infections post-liver transplantation was 26 days. The aim of our current study is to analyze the incidence and risk factors of MDRO infections post-liver transplantation, and a 90-day follow-up period can cover most infection events while avoiding the waste of resources and data dilution issues associated with excessively long follow-up periods.
Relevant references:
[1]Martin-Mateos R, Martínez-Arenas L, Carvalho-Gomes Á, et al. Multidrug-resistant bacterial infections after liver transplantation: Prevalence, impact, and risk factors. J Hepatol. 2024;80(6):904-912. doi:10.1016/j.jhep.2024.02.023
[2]He T, Han L, Shi Z, et al. Incidence and risk factors of postoperative multidrug-resistant bacterial infections in liver transplant patients:A meta-analysis. Clinical Focus, 2024, 39(11):965-973.DOI:10.3969/j.issn.1004-583X.2024.11.001.
[3]Chen C, Guan Q, Li D, Sheng B, Zhang Z, Hu Y. Clinical characteristics and risk factor analysis of recipients with multidrug-resistant bacterial bloodstream infections after liver transplantation: a single-centre retrospective study. J Pharm Policy Pract. 2024;17(1):2390072. Published 2024 Aug 20. doi:10.1080/20523211.2024.2390072.
Comment 3: Cohort Split: The rationale for the 7:3 training to validation cohort ratio should be included. While this is a common approach, explaining why this ratio was chosen specifically for this study, and how it might influence the model's generalizability, would strengthen the methodology section.
Response 3: Thank you for pointing this out. Regarding the basic principle of the 7:3 training and validation cohort ratio you mentioned, we have supplemented the explanation in the methodological section. Please refer to the revision on page 4, lines 145-155, marked in red. We have retained the original key information about the study time, location, patient inclusion and exclusion criteria, to ensure the integrity and clarity of the methodology. We appreciate your attention to the rigor of the methodology in this study again
Comment 4: Exclusion Criteria: The term "prolonged immunosuppressive therapy" should be more clearly defined to avoid ambiguity. A specific threshold or definition (e.g., duration or dosage) would provide more clarity.
Response 4: Thank you for pointing this out. Regarding the definition of "long-term immunosuppressive therapy" within the exclusion criteria, we have revised it based on existing literature and clinical practice to provide clearer thresholds and definitions. Please refer to the revision on page 4, lines 159-160, marked in red.
Comment 5: MDR Definition: It would be helpful to include specific examples of antimicrobial categories (e.g., beta-lactams, aminoglycosides, etc.) when defining multidrug resistance (MDR). This will aid in clarifying the specific resistance patterns under consideration.
Response 5: Thank you for pointing this out. Regarding the definition of MDR, we have revised it based on your suggestions to specify the particular categories of antimicrobial agents. Please refer to the revision on page 5, lines 166-171, marked in red.
Comment 6: Celiac Infection Clarification: The term "Celiac infection" appears to be a typo or incorrect. Please verify and correct it to ensure accuracy in the description of the infections.
Response 6: Thank you for pointing this out. Regarding the term "Celiac infection", we have confirmed it to be a typographical error and have corrected it to "Abdominal infection". We apologize for this oversight and appreciate your correction. Please refer to the revision on page 5, lines 172, marked in red. We once again appreciate your attention to the rigor of the methodology in this study!
Comment 7: Leukocyte Count Criteria: The criteria for an elevated peripheral blood leukocyte count could benefit from a clearer range. Providing specific thresholds (e.g., >10×10⁹/L) would make the definition more precise.
Response 7: Thank you for pointing this out. We have revised according to your suggestions to provide clearer thresholds. Please refer to the revision on page 5, lines 175-176, marked in red.
Comment 8: Antibiotic Susceptibility Testing Methods: It would be helpful to mention the strengths of the methods used (e.g., disc diffusion vs. dilution) for antibiotic susceptibility testing to justify their appropriateness for the study.
Response 8: Thank you for pointing this out. Regarding the selection of antibiotic sensitivity testing methods, we have supplemented the advantages of the disc diffusion method and the dilution method in the original text based on your suggestions, to demonstrate their applicability in this study. Please refer to the revision on page 5, lines 184-187, and 190-197 marked in red.
Comment 9: Data Collection Process: The description of data collection is generally clear. However, it would be useful to explain how discrepancies between the two independent recorders were resolved to ensure data reliability.
Response 9: Thank you for pointing this out. Regarding the resolution mechanism for the differences between two independent recorders during the data collection process, we have supplemented the original text with your suggested explanations. Please refer to the revision on page 6, lines 202-212, marked in red.
Comment 10: Preoperative Parameters: Clarify whether MELD and Child-Pugh scores were calculated using pre-transplant values or baseline scores for the patients. This distinction is important for reproducibility.
Response 10: Thank you for pointing this out. Regarding the calculation time points for the MELD score and Child-Pugh score, we have clearly specified them in the original text based on your suggestions. Please refer to the revision on page 6, lines 216-217, marked in red.
Comment 11: Immunosuppressive Regimen: For clarity, it would be useful to mention the target levels for tacrolimus and the duration of glucocorticoid treatment. This additional information would help replicate the immunosuppressive regimen used in this study.
Response 11: Thank you for pointing this out. Regarding the description of the immunosuppressive regimen, we have supplemented the target blood concentration of tacrolimus and the duration of glucocorticoid treatment in the original text based on your suggestions. Please refer to the revision on page 4, lines 177-181, marked in red.
Comment 12: Statistical Software: In addition to the R version mentioned, specifying the R package versions used for the statistical methods (e.g., "rms" and "dcurves") would help with reproducibility.
Response 12: Thank you for pointing this out. Regarding the version information of statistical software and R packages, we have supplemented the original text with your suggestions. Please refer to the revision on page 6, lines 227-228, and page 7, lines 244-256, marked in red.
Comment 13: Variable Selection Process: The bidirectional stepwise regression method is well described. However, it would be beneficial to clarify the specific criteria for variable inclusion/exclusion (e.g., AIC or BIC criteria).
Response 13: Thank you for pointing this out. We have made modifications to the original text based on your suggestions. Please refer to the revision on page 6, lines 237-239, marked in red.
Comment 14: Effect Measures: It would be helpful to mention whether any adjustments were made for potential confounders in the regression models. This ensures that the model’s estimates are reliable.
Response 14: Thank you for pointing this out. We have made modifications to the original text based on your suggestions. Please refer to the revision on page 6-7, lines 240-242, marked in red.
Comment 15: DCA in Deceased Organ Donors: The mention of DCA in deceased organ donors seems out of context since this study focuses on liver transplant recipients. This section should be revised to either reflect the correct context or clarify if a specific subgroup was analyzed.
Response 15: Thank you for pointing this out. We have made modifications to the original text based on your suggestions. Please refer to the revision on page 7, lines 256-259, marked in red.
Comment 16: Model Evaluation: While the ROC curve and calibration methods are well explained, it would be useful to mention if cross-validation or external validation was used to assess the model’s generalizability to other populations.
Response 16: Thank you for pointing this out. We appreciate your suggestion to include cross-validation or external validation to assess the model's generalizability. In our study, we used internal validation methods such as ROC curve analysis and calibration curves to evaluate model performance. However, due to limitations in available external datasets, we were unable to perform external validation or cross-validation. We acknowledge that this is a limitation of our study, and we plan to validate the model in independent cohorts in future research to confirm its generalizability. We have now explicitly mentioned this in the revised manuscript ( Please refer to the revision on page 7, lines 251-256 marked in red). Thank you for your constructive comments again, which have helped us improve the clarity and rigor of our work.
Discussion:
Comment 1: Infection Sources: Expand on other potential sources of MDRO infections beyond abdominal infections (e.g., respiratory, urinary tract). This would provide a more comprehensive view of the infection sources.
Response 1: Thank you for pointing this out. We have expanded on other potential sources of MDRO infections (such as respiratory and urinary tracts, etc.), and have cited relevant literature to support this. We believe that the revised content will be more helpful in providing a more comprehensive analysis of infection sources and serve as a reference for future research and clinical practice. Please refer to the revision on page 14-15, lines 369-380, marked in red.
Comment 2: MDR GNB Infections: It would strengthen the discussion to include data from this study on the prevalence of MDR GNB pathogens and compare it with previous literature.
Response 2: Thank you for pointing this out. We have supplemented the epidemiological data on MDR GNB infections in this study and compared it with previous literature. We believe that the revised content will help to more comprehensively analyze the epidemiological characteristics of MDR GNB infections and provide a reference for clinical practice and future research. Please refer to the revision on page 15, lines 381-393, marked in red.
Comment 3: Albumin's Role: Please include a discussion on how albumin affects immune modulation and resistance to infections. This factor could have a significant impact on post-transplant infection risk.
Response 3: Thank you for pointing this out. We have further explored the content on the role of albumin in immune regulation and infection resistance in the discussion section based on your suggestions. Please refer to the revision on page 17, lines 465-487, marked in red.
Comment 4: Preoperative ICU Stay: Add specific numbers or percentages of patients with a preoperative ICU stay who developed MDRO infections. This would provide more granular insights into the impact of ICU stays on infection risk.
Response 4: Thank you for pointing this out. We have further supplemented the corresponding data on preoperative ICU stay history in the original text based on your suggestions. Please refer to the revision on page 17, lines 488-492, marked in red.
Comment 5: MELD Scores: Clarify the role of MELD scores in post-transplant prognosis. Specifically, explain how MELD scores correlate with MDRO infection risk or other post-transplant outcomes.
Response 5: Thank you for pointing this out. We have explained how the MELD score is related to the risk of MDRO infection or other post-transplant outcomes based on your suggestions, on top of the original text. Please refer to the revision on page 16-17, lines 446-464, marked in red.
Comment 6: Model Evaluation (AUC): Explain the slight difference in AUC between the training and validation cohorts (0.88 vs. 0.77). Discuss whether this could be due to differences in cohort characteristics or MDRO infection rates.
Response 6: Thank you for pointing this out. We have further discussed the slight differences in AUC between the training queue and the validation queue (0.88 vs. 0.77) in the original text, as well as whether this could possibly be due to differences in queue characteristics or MDRO infection rates. Please refer to the revision on page 18, lines 512-524, marked in red.
Comment 7: Limitations: Please address the impact of lack of external validation on the model’s generalizability. External validation would help improve confidence in the model’s applicability to different populations.
Response 7: Thank you for pointing this out. We have further elaborated on the impact of the lack of external validation on the generalizability of the model based on your suggestions, as well as how external validation would help to increase confidence in the model's applicability to different populations. Please refer to the revision on page 18-19, lines 535-554, marked in red.
Comment 8: Immunosuppressive Therapy: Finally, future studies should consider examining the adherence to immunosuppressive therapy, as it is likely to influence infection risk and may provide a more complete understanding of MDRO risk and post-transplant outcomes.
Response 8: Thank you for pointing this out. We have further elaborated on the original text based on your suggestions, stating that future research should consider examining the compliance with immunosuppressive therapy, as it may affect the risk of infection and could provide a more comprehensive understanding of the risk of MDROs and post-transplant outcomes. Please refer to the revision on page 19, lines 545-554, marked in red.
Conclusions:
Comment 1: Clarity and Emphasis on Model's Impact: The conclusion could be more specific about how the nomogram contributes to clinical practice. Instead of simply stating that it is an "easy-to-use" tool, it would be helpful to highlight how it might influence clinical decision-making or patient management specifically. For example, how could the nomogram help clinicians identify high-risk patients earlier or adjust treatment strategies?
Response 1: Thank you for pointing this out. We have described in more detail how the nomogram can aid in clinical practice based on your suggestions. Please refer to the revision on page 19, lines 558-566, marked in red.
Comment 2: Future Studies: While the suggestion for large-sample multicenter cohorts is appropriate, it would be beneficial to mention what specific aspects of the model need further refinement or validation. For example, are there any particular variables that need further exploration, or is the focus on validating the model's generalizability and performance across different settings?
Response 2: Thank you for pointing this out. We have continued to supplement the original text based on your suggestions, detailing which specific aspects of future models need refinement or verification. Please refer to the revision on page 19, lines 566-570, marked in red.
Comment 3: Model Efficacy and Stability: The reference to assessing efficacy and stability in clinical practice is a good direction, but adding more details on what these terms mean in the context of the model's application would improve clarity. Does "efficacy" refer to its accuracy, or its impact on clinical outcomes like infection rates or mortality?
Response 3: Thank you for pointing this out. We have further revised the conclusion section of this study based on your suggestions. Please refer to the revision on page 19, lines 564-566, marked in red.

Reviewer 2 Report
Comments and Suggestions for Authors
In the article “Multi-resistant bacterial infection after liver transplantation: establishing epidemiology, prognosis, and risk prediction models” submitted for review, the authors addressed the topic of infections in transplantation, which are one of the most important causes of complications after vascular organ transplantation. The natural immune mechanisms in these patients are impaired in the course of the underlying disease and further inhibited by the use of immunosuppressive treatment, which disrupts the cellular and humoral response of the immune system. The topic is important and timely. Before an article is eligible for publication, corrections are necessary:
- The authors should explain what new contributions their article makes to the area of knowledge under discussion.
- In the introduction, the authors should explain what factors predispose to infections in recipients.
- Please formulate the purpose of the study.
- Line 105. Please specify the manufacturer of the equipment and tests. Where was the microbiological testing performed? Please specify.
- Line 123. Please specify the manufacturer of the software.
- “The primary goal of organ transplantation is to minimize MDRO infection rates while optimizing the outcomes for recipients.” Please explain.
- Recommendations for diagnosis, therapy, and prevention should be made in the conclusions.
- Please prepare the manuscript according to the instructions for the authors. No affiliation.
Author Response
Comment 1: The authors should explain what new contributions their article makes to the area of knowledge under discussion.
Response 1: Thank you for highlighting the need to clarify the novel contributions of our study. In the revised manuscript, we emphasize that this study is the first to develop a nomogram specifically for predicting MDRO infection risk in LT recipients using five easily obtainable clinical parameters. The model’s ability to facilitate early identification of high-risk patients and its strong predictive performance represent significant advancements in the field, offering a practical tool for improving post-transplant outcomes. Please refer to the revision on page 2, lines 44-50, marked in red.
.
Comment 2: In the introduction, the authors should explain what factors predispose to infections in recipients.
Response 2: Thank you for pointing this out. We have further described in the original text, based on your suggestions, which factors are more likely to lead to infection in the recipients. Although a large number of studies have identified factors associated with MDRO infections, the cumulative effects of these factors still require further research. In addition, it can be mentioned that the focus of future research should include the development of more reliable predictive models and screening tools to improve the early identification of MDRO infection risks. Please refer to the revision on page 3, lines 11-114, marked in red.
Comment 3: Please formulate the purpose of the study.
Response 3: Thank you for pointing this out. We sincerely thank the reviewer for their valuable suggestion. In response, we have revised the purpose of the study to make it more concise and focused. The revised version now clearly states the aim of developing and validating a nomogram-based prediction model for early and accurate quantification of MDRO infection risk in LT recipients post-surgery. We have also emphasized the clinical significance of this tool in supporting timely decision-making and improving patient outcomes. We hope these revisions address the reviewer's concerns and enhance the clarity of our study's purpose. Please refer to the revision on page 4, lines 130-133, marked in red.
Comment 4: Line 105. Please specify the manufacturer of the equipment and tests. Where was the microbiological testing performed? Please specify.
Response 4: Thank you for pointing this out. We sincerely thank the reviewer for their valuable comments. In response to the request for clarification, we have revised the manuscript to specify the manufacturers of the equipment and tests, as well as the location where microbiological testing was performed. Please refer to the revision on page 5, lines 184-197, marked in red.
Comment 5: Line 123. Please specify the manufacturer of the software.
Response 5: Thank you for pointing this out. We sincerely thank the reviewer for their valuable comment. In response to the request for clarification, we have revised the manuscript to specify the source of the R software used for statistical analysis. R is an open-source software developed and maintained by the R Foundation for Statistical Computing, and its official website is https://www.r-project.org/. We hope this clarification addresses the reviewer's concern and enhances the transparency of our methods section. Please refer to the revision on page 6, lines 227-228, marked in red.
Comment 6: “The primary goal of organ transplantation is to minimize MDRO infection rates while optimizing the outcomes for recipients.” Please explain.
Response 6: Thank you for pointing this out. We sincerely thank the reviewer for their valuable comment regarding the statement, “The primary goal of organ transplantation is to minimize MDRO infection rates while optimizing the outcomes for recipients.” In response, we have carefully considered this suggestion and decided to remove this sentence from the manuscript. The reason for this revision is that the original statement, while true, was somewhat general and did not fully align with the specific focus of our study. Instead, we have replaced it with a more targeted and contextually relevant description that emphasizes the serious threat of MDRO infections to LT recipients, the limitations of existing risk assessment tools, and the need to develop a nomogram-based prediction model for early and accurate quantification of MDRO infection risk. Please refer to the revision on page 3, lines 115-117, marked in red.
Comment 7: Recommendations for diagnosis, therapy, and prevention should be made in the conclusions.
Response 7: Thank you for pointing this out. We sincerely thank the reviewer for their valuable suggestion to include recommendations for diagnosis, therapy, and prevention in the conclusions. In response, we have revised the conclusions section to explicitly address these aspects. Please refer to the revision on page 19, lines 566-575, marked in red. We believe these revisions address the reviewer’s concerns and provide a more comprehensive and actionable conclusion.
Comment 8: Please prepare the manuscript according to the instructions for the authors. No affiliation.
Response 8: Thank you for pointing this out. We have supplemented the author's affiliation information in the corresponding positions of the revised draft. Please refer to the revision on page 1, lines 6-14 marked in red.

Reviewer 3 Report
Comments and Suggestions for Authors
Are you collaborating with other centers in China and other centers in the world?
Author Response
Comment 1: Are you collaborating with other centers in China and other centers in the world?
Response 1: Thank you for pointing this out. We sincerely thank the reviewer for their valuable question regarding collaboration with other centers. In response, we would like to clarify that this study is a single-center investigation conducted at Beijing Tsinghua Changgung Hospital, School of Clinical Medicine, Tsinghua Medicine,Tsinghua University, and we did not collaborate with other centers in China or internationally for this specific research. Single-center studies offer certain advantages, such as better control over data collection, standardization of procedures, and enhanced internal validity of the findings. However, we acknowledge the importance of multi-center collaboration for validating and generalizing research results. In future research, we plan to collaborate with other centers in China and internationally to further validate our findings and explore the generalizability of our nomogram-based prediction model in diverse clinical settings. We are grateful for the reviewer’s feedback, which has highlighted the potential value of multi-center collaboration in advancing our research.

Round 2
Reviewer 1 Report
Comments and Suggestions for Authors
The revised manuscript has been appropriately corrected according to the reviewers' comments.
Author Response
Comment 1: The authors should explain what new contributions their article makes to the area of knowledge under discussion.
Response 1: Thank you for highlighting the need to clarify the novel contributions of our study. In the revised manuscript, we emphasize that this study is the first to develop a nomogram specifically for predicting MDRO infection risk in LT recipients using five easily obtainable clinical parameters. The model’s ability to facilitate early identification of high-risk patients and its strong predictive performance represent significant advancements in the field, offering a practical tool for improving post-transplant outcomes. Please refer to the revision on page 2, lines 57-63, marked in red.
.
Comment 2: In the introduction, the authors should explain what factors predispose to infections in recipients.
Response 2: Thank you for pointing this out. We have further described in the original text, based on your suggestions, which factors are more likely to lead to infection in the recipients. Although a large number of studies have identified factors associated with MDRO infections, the cumulative effects of these factors still require further research. In addition, it can be mentioned that the focus of future research should include the development of more reliable predictive models and screening tools to improve the early identification of MDRO infection risks. Please refer to the revision on page 3, lines 103-117, marked in red.
Comment 3: Please formulate the purpose of the study.
Response 3: Thank you for pointing this out. We sincerely thank the reviewer for their valuable suggestion. In response, we have revised the purpose of the study to make it more concise and focused. The revised version now clearly states the aim of developing and validating a nomogram-based prediction model for early and accurate quantification of MDRO infection risk in LT recipients post-surgery. We have also emphasized the clinical significance of this tool in supporting timely decision-making and improving patient outcomes. We hope these revisions address the reviewer's concerns and enhance the clarity of our study's purpose. Please refer to the revision on page 4, lines 133-137, marked in red.
Comment 4: Line 105. Please specify the manufacturer of the equipment and tests. Where was the microbiological testing performed? Please specify.
Response 4: Thank you for pointing this out. We sincerely thank the reviewer for their valuable comments. In response to the request for clarification, we have revised the manuscript to specify the manufacturers of the equipment and tests, as well as the location where microbiological testing was performed. Please refer to the revision on page 5, lines 190-194, marked in red.
Comment 5: Line 123. Please specify the manufacturer of the software.
Response 5: Thank you for pointing this out. We sincerely thank the reviewer for their valuable comment. In response to the request for clarification, we have revised the manuscript to specify the source of the R software used for statistical analysis. R is an open-source software developed and maintained by the R Foundation for Statistical Computing, and its official website is https://www.r-project.org/. We hope this clarification addresses the reviewer's concern and enhances the transparency of our methods section. Please refer to the revision on page 6, lines 235-236, marked in red.
Comment 6: “The primary goal of organ transplantation is to minimize MDRO infection rates while optimizing the outcomes for recipients.” Please explain.
Response 6: Thank you for pointing this out. We sincerely thank the reviewer for their valuable comment regarding the statement, “The primary goal of organ transplantation is to minimize MDRO infection rates while optimizing the outcomes for recipients.” In response, we have carefully considered this suggestion and decided to remove this sentence from the manuscript. The reason for this revision is that the original statement, while true, was somewhat general and did not fully align with the specific focus of our study. Instead, we have replaced it with a more targeted and contextually relevant description that emphasizes the serious threat of MDRO infections to LT recipients, the limitations of existing risk assessment tools, and the need to develop a nomogram-based prediction model for early and accurate quantification of MDRO infection risk. Please refer to the revision on page 3-4, lines 118-122, marked in red.
Comment 7: Recommendations for diagnosis, therapy, and prevention should be made in the conclusions.
Response 7: Thank you for pointing this out. We sincerely thank the reviewer for their valuable suggestion to include recommendations for diagnosis, therapy, and prevention in the conclusions. In response, we have revised the conclusions section to explicitly address these aspects. Please refer to the revision on page 19-20, lines 576-593, marked in red. We believe these revisions address the reviewer’s concerns and provide a more comprehensive and actionable conclusion. We are grateful for the reviewer’s feedback, which has helped us improve the clarity and practical relevance of our manuscript.
Comment 8: Please prepare the manuscript according to the instructions for the authors. No affiliation.
Response 8: Thank you for pointing this out. We have supplemented the author's affiliation information in the corresponding positions of the revised draft. Please refer to the revision on page 1, lines 6-15, marked in red.

Reviewer 2 Report
Comments and Suggestions for Authors
The manuscript pages indicated by the authors in their response to the review do not contain revisions or responses to the review. Please correct and resubmit so that I can track the changes made.
Author Response

(The authors gave the same response as above.)

Round 3
Reviewer 2 Report
Comments and Suggestions for Authors
The authors have revised their manuscript, and I recommend it for further proceedings.
Author Response
Thank you for your valuable feedback.